# The Impact of the COVID-19 Pandemic and Lockdowns on Self-Poisoning and Suicide in Sri Lanka: An Interrupted Time Series Analysis

**DOI:** 10.3390/ijerph20031833

**Published:** 2023-01-19

**Authors:** Thilini Rajapakse, Tharuka Silva, Nirosha Madhuwanthi Hettiarachchi, David Gunnell, Chris Metcalfe, Matthew J. Spittal, Duleeka Knipe

**Affiliations:** 1Department of Psychiatry, Faculty of Medicine, University of Peradeniya, Peradeniya 20400, Sri Lanka; ushanitharuka@yahoo.com; 2South Asian Clinical Toxicology Research Collaboration, Faculty of Medicine, University of Peradeniya, Peradeniya 20400, Sri Lanka; 3Toxicology Unit, Teaching Hospital, Peradeniya 20400, Sri Lanka; niro_madhu@yahoo.com.sg; 4Population Health Sciences, Bristol Medical School, University of Bristol, Bristol BS8 2PS, UK; d.j.gunnell@bristol.ac.uk (D.G.); chris.metcalfe@bristol.ac.uk (C.M.); 5National Institute of Health Research Biomedical Research Centre, Oakfield House, Oakfield Grove, Bristol BS8 2BN, UK; 6Centre for Mental Health, Melbourne School of Population and Global Health, University of Melbourne, Melbourne, VIC 3010, Australia; m.spittal@unimelb.edu.au

**Keywords:** suicide, self-harm, Sri Lanka, COVID-19, pandemic, low- and middle-income country, global mental health

## Abstract

Evidence from high-income countries suggests that the impact of COVID-19 on suicide and self-harm has been limited, but evidence from low- and middle-income countries is lacking. Using data from a hospital-based self-poisoning register (January 2019–December 2021) and data from national records (2016–2021) of suicide in Sri Lanka, we aimed to assess the impact of the pandemic on both self-poisoning and suicide. We examined changes in admissions for self-poisoning and suicide using interrupted time series (ITS) analysis. For the self-poisoning hospital admission ITS models, we defined the lockdown periods as follows: (i) pre-lockdown: 01/01/2019–19/03/2020; (ii) first lockdown: 20/03/2020–27/06/2020; (iii) post-first lockdown: 28/06/2020–11/05/2021; (iv) second lockdown: 12/05/2021–21/06/2021; and (v) post-second lockdown: 22/06/2021–31/12/2021. For suicide, we defined the intervention according to the pandemic period. We found that during lockdown periods, there was a reduction in hospital admissions for self-poisoning, with evidence that admission following self-poisoning remained lower during the pandemic than would be expected based on pre-pandemic trends. In contrast, there was no evidence that the rate of suicide in the pandemic period differed from that which would be expected. As the long-term socioeconomic impacts of the pandemic are realised, it will be important to track rates of self-harm and suicide in LMICs to inform prevention.

## 1. Background

The COVID-19 pandemic has led to substantial disruption around the world. Social isolation, financial strain, fears of contracting COVID-19, and the illness itself have all been linked to increased symptoms of anxiety, depression, and suicidal ideation in the community [1,2]. Yet the available evidence indicates that suicide rates did not increase during the early part of the pandemic [2,3,4,5,6] in most high-income countries. Initial evidence from Sri Lanka found that the number of people presenting to hospital due to self-harm by self-poisoning in the first five months of the pandemic was lower when compared to expected trends [7]. Similar reductions in hospital presentations due to self-harm in the early part of the pandemic have been reported internationally and may have reflected difficulties in accessing healthcare services during the pandemic or the use of alternate sources of support [8].

As the world emerges from the acute phase of the pandemic, new concerns arise about the long-term psychosocial sequalae, and the impact on rates of self-harm and suicide. What research there is on this topic has largely been conducted in high-income countries. Less is known about the psychosocial impact of the pandemic on low–middle-income countries (LMICs). Sri Lanka is one such country and had the second-highest rate of suicide in the world in the mid-1990s [9]. This high rate has declined since then, largely due to falls in pesticide suicides due to bans on the sale of the most toxic products. However, suicide (now mostly by hanging) and self-harm by self-poisoning remain significant public health problems in the country [10,11,12]. Sri Lanka currently faces very significant economic problems, with increasing unemployment and greater numbers of people experiencing poverty [13,14]. Other LMICs face similar economic challenges [15]. Given these ongoing post-COVID socio-economic and other stressors, there are significant concerns that the initial fall in self-harm hospital admissions seen in the early part of the pandemic in Sri Lanka, may now be reversed, with possible rises in rates of suicide and self-harm in more recent months. Therefore, the objectives of this study were three-fold: (i) to explore if the initial reduction in hospital presentation for self-harm (by self-poisoning) was sustained once lockdown measures in Sri Lanka were rescinded and whether similar reductions were observed in subsequent lockdowns; (ii) to explore whether the impact of the pandemic on self-poisoning differed by sex, age, and type of poison ingested; and (iii) to describe rates of suicide in the country during the period 2016–2021. These findings are important for setting priorities for suicide prevention strategies in Sri Lanka and are likely to be informative for other South Asian and LMICs as well.

## 2. Methods

### 2.1. Setting

This study is based in Sri Lanka, a country with a population of 21 million (Census 2011). During the initial phases of the pandemic, there was a range of public health prevention measures introduced in Sri Lanka, including several nationwide lockdowns. The first national lockdown was introduced on 20 March 2020 and took the form of an island-wide continuous police curfew (i.e., all individuals had to strictly remain indoors). This was continued until late April, with brief periods when the curfew was lifted for a few hours on selected days and times, to allow people to leave their homes. Thereafter, there was a gradual reduction in periods of travel restriction, and on 28 June 2020, all national lockdown measures were lifted. Following this, there were public health measures to curb the spread of the virus, but travel restrictions were much less stringent than before. The second wave of COVID-19 in Sri Lanka started in October 2020. During the second wave, a quarantine curfew was imposed within the Western province only (population 2.3 million; 11% of the population of Sri Lanka), with people not being able to leave the Western province without permission, but these travel restrictions were much less strict than during the first wave. There was no quarantine curfew in the Central province or other parts of the country at the time, although in November–December 2020, a few small, localized areas in the Kandy district were isolated from time to time, which may have affected the patient population in the current study.

In the first quarter of 2021, most travel restrictions were removed. However, after the Sri Lankan New Year celebrations in April 2021, there was an additional wave of infections, which led to a second national lockdown from 12 May 2021 onwards. During this period, interprovincial travel was limited, but restrictions in travel were much less stringent compared to the initial curfew/lockdown, and there was no police curfew. This continued until 21 June 2021. With the resurgence of the new delta variant, interprovincial travel restrictions were imposed from August to late October 2021, but other than that, travel restrictions during this period were minimal. Safety measures were continued until the end of the year to prevent the spread of the virus.

During the above periods of island-wide lockdown and periods of travel restriction (2nd and 3rd wave), the government also restricted all legal sales of alcohol. However, anecdotal reports suggest that illicit alcohol production increased during this period.

Two sets of data were used for this analysis, one pertaining to hospital admissions for self-poisoning and the other to suicide deaths. The self-poisoning data used for this analysis were collected from the Teaching Hospital Peradeniya (THP) in the Kandy district in the Central province of Sri Lanka. The Kandy district mirrors the demographic profile of the country in terms of age and sex distribution, with a Sinhalese and Buddhist majority. National suicide data were obtained from the Department of Police, Division of Statistics, Sri Lanka.

### 2.2. Self-Poisoning Hospital Data

In 2020, a register was established to track the impact of COVID-19 on hospital admission for self-poisoning in Sri Lanka [7]. Data prior to the establishment of the register in June 2020 were collected retrospectively from medical records (referred to as Bed Head Tickets or BHTs) and the ward admission book and included data from January 2019. From June 2020 onwards, we have collected data on all persons presenting to THP for medical management of self-poisoning. All persons presenting to THP due to self-poisoning are admitted to the toxicology ward for medical treatment, and there were no changes to self-poisoning admission procedures due to the pandemic. During data collection, patients admitted due to self-poisoning were identified using the ward admission book and additional data were collected using BHTs. Using a standard extraction sheet, data were gathered from medical records for all persons admitted due to self-poisoning to the ward between January 2019 and December 2021. Data on date of admission, age/date of birth, sex, and type of poison ingested were collected.

### 2.3. Suicide Data

National suicide counts by age, sex, and method were obtained from the Department of Police, Division of Statistics, between 2016 and 2021. Data for 2016–2017 were available in a monthly format, whereas 2018–2021 data were available as cumulative quarterly data.

### 2.4. Data Management

All data cleaning and analysis were conducted in line with a pre-specified published analysis plan [16]. At the time of publishing our analysis plan, we only had self-poisoning data up to April 2021 and suicide data to 2020. The original analysis plan was restricted to assessing whether there was a temporary step change (i.e., level change) in the number of hospital admissions for self-poisoning during the lockdown period, after which the underlying trend would return to pre-pandemic levels. With the availability of additional data, we modified our original analysis plan to explore the impact of the second lockdown on self-poisoning rates.

### 2.5. Self-Poisoning Hospital Data

We included all patients who had a date of admission between January 2019 and December 2021 (date of data extraction: 31 May 2022). Patient age at admission was calculated using their date of birth (where available) or reported age. Consistent with our previous analysis of the same data, we categorised patients into young (<25-year-olds) and older (25+) individuals. We categorised poisoning into medicinal, agrochemicals, or plant/other based on the toxicological agent. Data were converted to weekly counts of hospital admissions for all people and stratified by age, sex, and method of poisoning. Weeks were coded to begin on Sunday and end on Saturday.

### 2.6. Suicide Data

The monthly suicide data for the years 2016 and 2017 were grouped into quarters, and quarterly suicide counts were calculated from the cumulative quarterly suicide data available for 2018–2021. We categorised age into four groups (8–25, 26–35, 36–55, and 55+), which reflect the groups which have the most similar age-specific suicide trends, and the categories which have been used in the previous analysis of these data [11]. The method of suicide was categorised into three main groups (hanging, pesticide poisoning, and other methods).

### 2.7. Statistical Analyses

We graphically present weekly and quarterly changes in self-poisoning hospital admissions (January 2019–December 2021) and suicide (2016–2021). We present this by sex, age, and methods, as well as overall trends.

We examined changes in admissions for self-poisoning and suicide using interrupted time series (ITS) analysis [17]. Given the different time periods and frequency of data available for self-poisoning admission and suicide deaths, we had to define the pandemic/lockdown periods for each dataset slightly differently. For the self-poisoning hospital admission ITS models, we defined the lockdown periods as follows: (i) pre-lockdown: 1 January 2019–19 March 2020; (ii) first lockdown: 20 March 2020–27 June 2020; (iii) post-first lockdown: 28 June 2020–11 May 2021; (iv) second lockdown: 12 May 2021–21 June 2021; and (v) post-second lockdown: 22 June 2021–31 December 2021. The quarterly nature of the suicide data meant that we defined the intervention according to the pandemic period rather than the lockdown (i.e., pre-pandemic: Q1 (January–March) 2016–Q1 2020; pandemic period: Q2 (April–June) 2020–Q4 (October–December) 2021).

We conducted a series of ITS analyses by fitting Poisson regression models with a scale parameter to account for overdispersion. For the self-poisoning admission data, our outcome was the weekly number of self-poisoning hospital admissions. We fitted models for the overall number of presentations and then by age group and sex and by poison method. For the suicide data, our outcome was the quarterly number of suicide deaths, and we fit models by the overall number of suicide deaths and by age group, sex, and method. We used the *fp* function in Stata statistical software (version 16.1, StataCorp, College Station, TX, USA, 2017) to assess the model which fitted the data best in terms of incorporating longer-term time-trends which might be non-linear. We also fitted models with and without terms to account for seasonality and tested using a likelihood ratio test to see which of these models was a better fit for the data. We opted for the most parsimonious model that captured the major time trends and variations in the observations. In our statistical plan, we hypothesised that there would be a temporary step change (i.e., level change) in weekly hospital presentations, and, therefore, all our models on the self-poisoning datasets included a categorically coded predictor in the model which represents the different periods of the lockdown detailed above. Most fitted models were linear and did not require a term to account for seasonality. The equation for the self-poisoning admission data without accounting for seasonality was: 


*ln(episodes in weeks) = β0 + β1(time)+ β2(1st lockdown weeks) + β3(post-1st lockdown weeks) + β4(2nd lockdown weeks) + β5(post-2nd lockdown weeks).*


Where *β1 (time*) takes into account the underlying trend prior to the pandemic starting, *β0* represents the baseline level at time=0, and *β2-5* is a binary variable which indicated whether the time period is one of the lockdown periods of interest. For the suicide outcome analysis, we only included one interruption term (*β2*) to indicate the pandemic period.

We tested for evidence of this temporary step change in overall self-poisoning hospital admissions by age and sex and by method. We report the effect of this predictor as a rate ratio (RR). If a temporary step change has occurred, the RR during the lockdown periods will be different (higher or lower) than pre-lockdown trends, but the post-lockdown period RRs will show no evidence of a difference. This would indicate a “bouncing back” effect. 

For the suicide data, we hypothesised that there would be a step change after the onset of the pandemic and included a binary-coded variable to indicate this in our models. We used the mid-year population estimates for 2016–2020 from the Registrar General’s Department, Sri Lanka, as an offset to calculate rates of suicide. These models were fit using the same approach used for the hospital presentation data. We fitted models on overall rates by age, sex, and method.

### 2.8. Ethical Approval

Ethical approval for the register was obtained from the Ethics Review Committee of the Faculty of Medicine, University of Peradeniya, Sri Lanka.

## 3. Results

### 3.1. Hospital Admissions Due to Self-Poisoning, Peradeniya

There were 2259 hospital presentations for self-poisoning between 1 January 2019 and 31 December 2021, and BHT data were missing for 3% of cases. Table 1 provides basic descriptive data of the study sample by period.

Figure 1 shows the overall number of hospital admissions for self-poisoning. The ITS indicates that during all lockdown (during and post) periods, there was a reduction in the number of hospital presentations for self-poisoning, which is over and above a steady long-term reduction in presentations following self-poisoning. The largest reductions in admissions were during the lockdown periods, where a 50% reduction was observed during both lockdowns (first lockdown: RR 0.54 95% CI 0.44, 0.67; second lockdown: RR 0.55 95% CI 0.37, 0.81) (Table 2). The number of admissions did not return to pre-pandemic levels after the lockdowns were lifted.

Figure 2 presents the rate of self-poisoning hospital admissions by sex and age group. During the first lockdown, the rate of hospital admissions for self-poisoning dropped in each sex and age stratum, and we observed a similar reduction in the second lockdown compared to pre-pandemic trends (Table 2). The model estimates when stratified by sex and age are consistent with a temporary step change in rates of self-poisoning admissions. There was also evidence of a temporary step change in the number of hospital admissions for medicinal poisoning during the first pandemic (Figure 3 and Table 2) but not the second. We also observed a reduction in the number of hospital presentations for plant and other poisonings during the period between the first and second lockdown (RR 0.55 95% CI 0.33, 0.91).

### 3.2. Suicide Numbers, Nationwide

A total of 18,979 people died by suicide between 2016 and 2021 in Sri Lanka. The crude rate of suicide in Sri Lanka in 2021 was 17.1 per 100,000 (95% CI 16.5, 17.8 per 100,000). The age and sex profile of deaths were similar during the pandemic periods (Table 3). Figure 4, Figure 5 and Figure 6 present the rate of suicide during the study period. The ITS models found limited statistical evidence of a step change in the rate of suicide during the pandemic compared to the pre-pandemic trends, with the exception of those aged 26–35 years. In this group, there was statistical evidence of a 26% reduction in the rate of suicide during the pandemic (RR 0.74 95% CI 0.54, 1.00). Suicide rates were consistently higher among males and more common in older compared to younger persons. Although the overall number of suicide deaths has remained relatively constant (mean 3163, SD 105), there was an upward trend of suicide by hanging, with the highest observed number of suicide deaths by hanging being at the end of 2021.

## 4. Discussion

This study provides data on the impact of the pandemic and associated lockdowns on self-poisoning and suicide in a lower–middle-income country. We found that during the lockdown periods, there was a reduction in hospital presentations for self-poisoning, with evidence that presentation to services following self-poisoning remain lower during the pandemic than would be expected based on pre-pandemic trends. This pattern was seen in all sub-group analyses (i.e., by sex, age, and method), although small numbers in particular strata meant that there was not always statistical evidence of a reduction. In contrast, there was no evidence that the rate of suicide in the pandemic period differed from that which would be expected based on pre-pandemic trends.

### 4.1. Hospital Presenting Self-Poisoning

The finding of a temporary decrease in the number of hospital presentations due to self-poisoning, during each lockdown period is similar to reported findings internationally, where lockdowns have been associated with reduced hospital presentations for self-harm [8]. Difficulty accessing services due to travel restrictions, limited access to means of self-harm (most often medication), increased numbers of family members within the home, and seeking alternate sources of support may have all contributed to the reduced number of presentations [8].

During the first lockdown, we also observed a significant drop in the number of people presenting with self-poisoning from medicinal overdoses (compared to other substance ingestion). The drop in medicinal overdoses in the first lockdown may go hand-in-hand with a greater reduction in the number of young people (<25 years) presenting to hospital due to self-poisoning compared to older persons – young people may have been less able to travel during strict lockdown periods, and previous evidence from Sri Lanka has indicated that medicinal overdoses are relatively more common in younger people, whereas pesticide ingestion is more often seen in older people [18].

There were concerns as to whether the reduced hospital presentations due to fears of contracting COVID-19, during the periods of lockdown hid ongoing self-harm within the community, which did not present to healthcare services. However, we found that after lockdowns were lifted, the levels of hospital admissions due to self-poisoning did not rise above pre-pandemic trends, with evidence that they were lower than levels before the onset of the pandemic. The reasons for this are unclear. This might reflect ongoing concerns over the risk of infection in a hospital, though we do not have any data to be able to confirm this. 

Prior to the pandemic starting, there is evidence of a gradual decline in self-poisoning that present to hospital. Again, the reasons for this are unknown, but it could be that self-poisoning was gradually becoming less common as a method of self-harm and that other methods were being adopted, e.g., such as self-cutting, which are less likely to present to services. Another more worrying potential possibility is that method switching in self-harm has been to potentially more lethal methods, such as hanging. Any method substitution to potentially more lethal methods, even by those attempting self-harm impulsively with short premeditation, is of serious concern and this possibility warrants further exploration. Sri Lanka, similar to many other LMICs, is currently facing significant socio-economic problems and worsening levels of poverty [15]. Socio-economic stressors, early school drop-out, and unemployment are risk factors for self-harm, especially for young people, and in this context, ongoing monitoring of rates of self-poisoning and self-harm is of importance [4,19,20].

### 4.2. Suicide

We found that the number of suicide deaths nationwide remained relatively constant between 2016 and 2021, with limited evidence of a step change in incidence during the pandemic compared to the pre-pandemic trends. This is in keeping with international work which reported no increase in suicide rates during the early phase of COVID-19, although our work reflects both the early and late part of the pandemic [3,21]. 

The most common method of suicide from 2016 onwards in Sri Lanka has been by hanging, with suicide being more common among older people and males, in keeping with trends of suicide worldwide [22]. This is a shift from the patterns of suicide seen in Sri Lanka in the 1990s and early 2000s when the country saw a very high rate of suicide mostly commonly by pesticide ingestion, and more commonly among young males [23]. – this occurred at a time when toxic pesticides were widely available in the community. Subsequently in the mid-1990s, national measures were taken to restrict the availability of toxic pesticides, and furthermore, the import of all pesticides was banned for a period during 2021 [10,24]. The gradual increase in the number of suicide deaths by hanging, seen throughout the study period is likely to reflect an underlying shift in method choice and some degree of method substitution [25]. 

Sri Lanka, akin to many other developing countries, is now facing an unprecedented economic crisis in the wake of the pandemic. Risk factors for suicide, such as socio-economic stressors, increasing levels of poverty and unemployment, and associated debt are already on the increase [4,19,26,27]. This is likely to go hand-in-hand with other triggers for self-harm and suicide, such as relationship strains, interpersonal conflict, depression, and increasing substance misuse [28,29,30]. Vulnerable groups, such as the elderly, young people, those below the poverty line, and those with psychiatric morbidity will be at increased risk [30]. Closely monitoring suicide and self-harm will be imperative going forward.

### 4.3. Strengths and Limitations

This is one of the very few studies investigating the impact of the pandemic on suicide and self-harm in populations living in LMICs [6] and presented data on both self-harm and suicide. It includes data from several years and is able, therefore, to model underlying trends. We conducted our analysis using a pre-registered analysis plan and have indicated when we have deviated from this. Despite these strengths, there are methodological limitations which need to be considered when interpreting the findings of this study. Firstly, the data for self-poisoning was obtained using hospital admissions to a large tertiary care hospital in the Central province of the country and, therefore, may not reflect self-harm in all parts of the country. Secondly, we only have access to data on people who presented to hospital after their self-harm, and we do not know whether self-harm in the community has changed during the pandemic period. Thirdly, data for rates of suicide were available in monthly or quarterly intervals and, therefore, we were unable to assess the impacts of lockdowns on suicide rates. Lastly, we were also not able to compare variations of suicide numbers by province, which is a limitation.

## 5. Conclusions

Despite concerns, available data internationally suggest that there has been no increase in the rates of suicide during the pandemic, and our findings from Sri Lanka are in keeping with this [3,6]. During the pandemic period of 2019–2021, admissions for self-poisoning showed an overall decreasing pattern, with temporary step reductions during the periods of lockdown. There was evidence that the number of hospital admissions for self-poisoning remained lower than would have been expected during non-lockdown pandemic periods. The overall rates of suicides nationwide during 2016–2021 did not show an increase during the pandemic. However, a significant red flag for concern is the upward trend of suicide by hanging, which appears to be independent of the pandemic. Similar suggestions of recently increased rates of suicide by hanging have also been reported from neighbouring India [31].

The negative socioeconomic consequences of the pandemic are starting to be realised in many countries. In this context, it is imperative to continue to closely monitor the rates of self-harm and suicide in Sri Lanka and other LMICs, which are the nations likely to experience the worst of these adversities. Taking universal, multi-pronged pre-emptive actions which go beyond the healthcare sector—such as measures for debt relief, job restructuring, and responsible media reporting of suicide—together with targeted identification and support for vulnerable groups, such as the elderly and those with psychiatric morbidity, may help in preventing an increase in suicide rates [1,2,27,32]. These strategies are worth exploring further and findings are likely to have implications for many LMICs.

## Figures and Tables

**Figure 1 ijerph-20-01833-f001:**
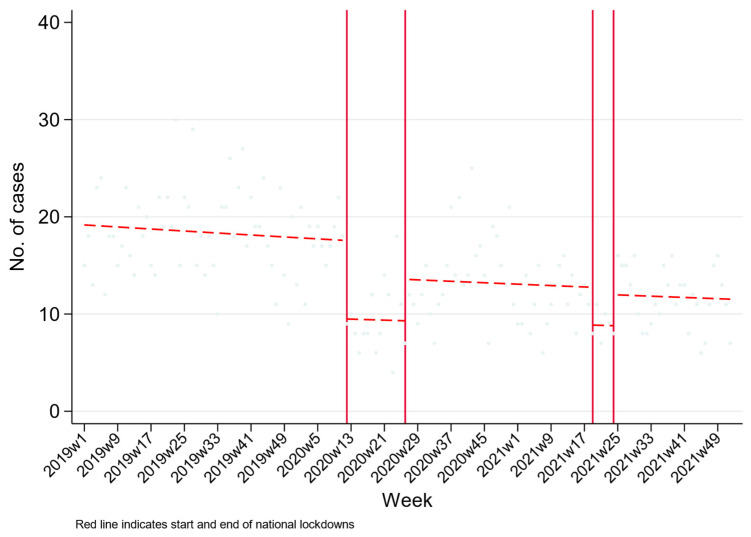
Changes in hospital presenting self-poisoning in Teaching Hospital Peradeniya, Sri Lanka, between January 2019 and December 2021.

**Figure 2 ijerph-20-01833-f002:**
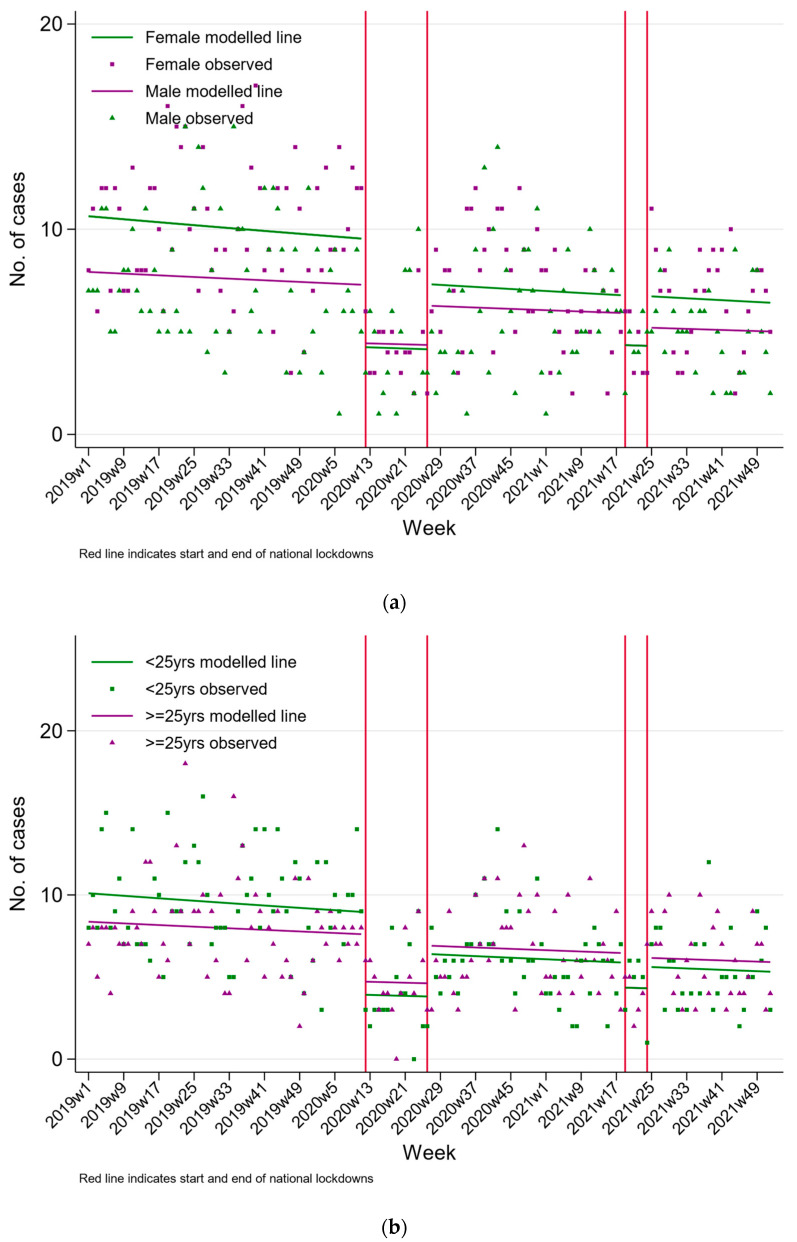
Changes in the rates of hospital presenting self-poisoning in Teaching Hospital Peradeniya, Sri Lanka, between January 2019 and December 2021 by (**a**) sex and (**b)** age group.

**Figure 3 ijerph-20-01833-f003:**
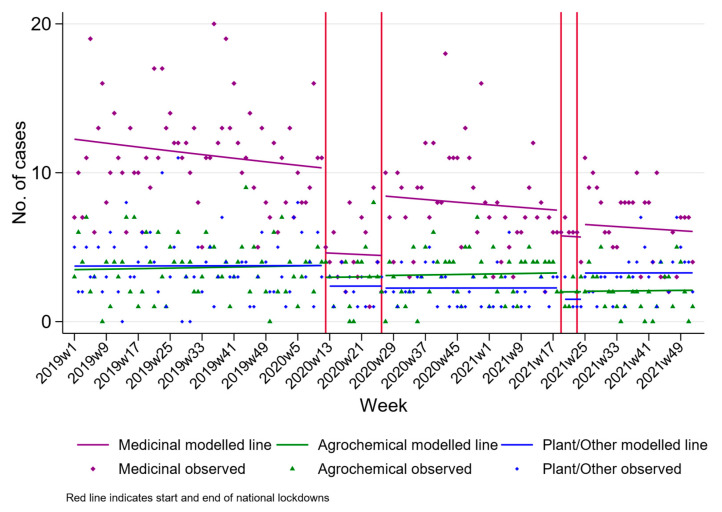
Changes in hospital presenting self-poisoning in Teaching Hospital Peradeniya, Sri Lanka, between January 2019 and December 2021 by poison type.

**Figure 4 ijerph-20-01833-f004:**
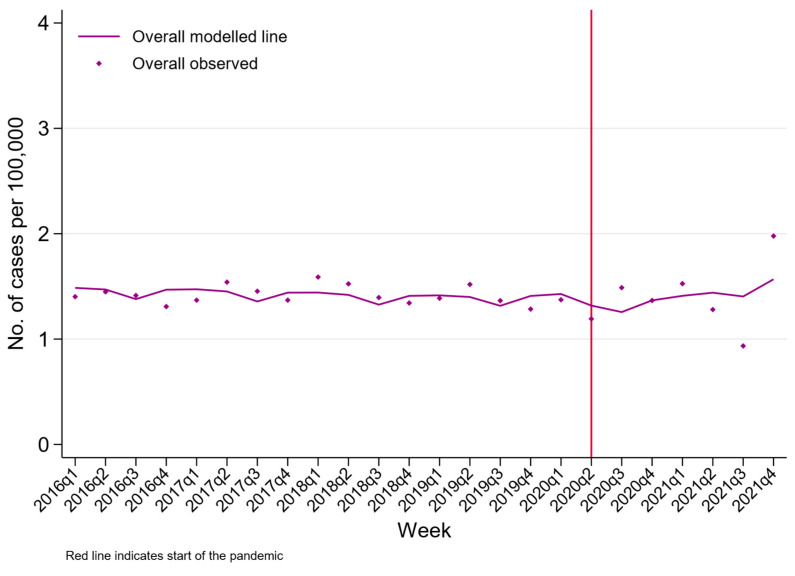
Changes in crude suicide rates in Sri Lanka between 2016 and 2021.

**Figure 5 ijerph-20-01833-f005:**
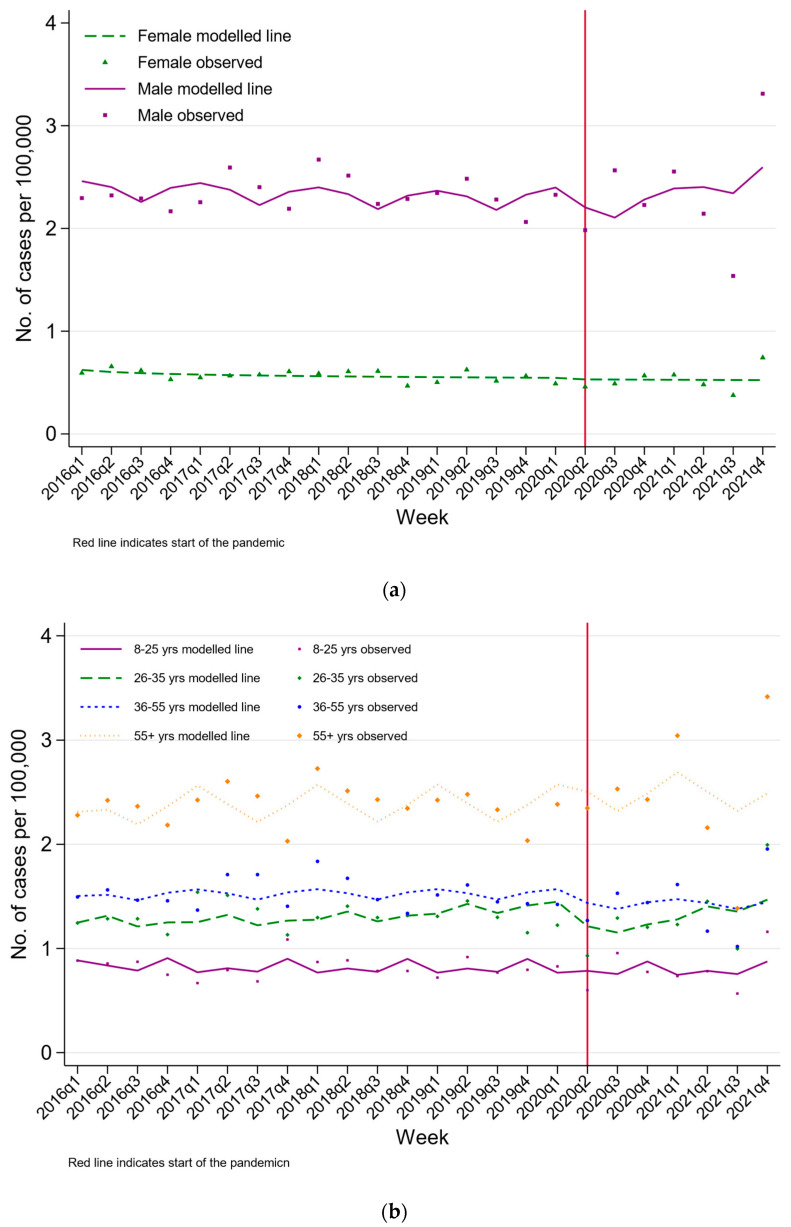
Changes in crude suicide rates in Sri Lanka between 2016 and 2021 by (**a**) sex and (**b**) age group.

**Figure 6 ijerph-20-01833-f006:**
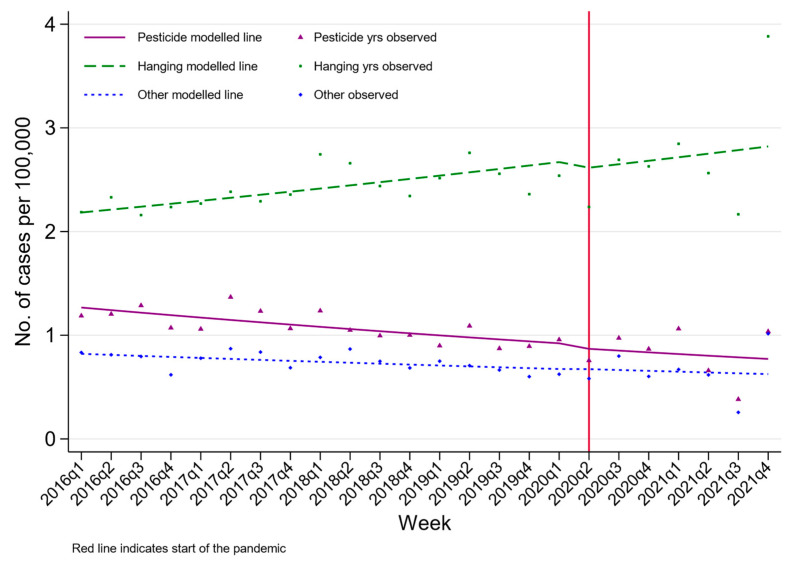
Changes in crude suicide rates in Sri Lanka between 2016 and 2021 by method.

**Table 1 ijerph-20-01833-t001:** Basic descriptive data on self-poisoning patients admitted to Teaching Hospital Peradeniya (January 2019–December 2021) by pandemic lockdown periods.

	Pandemic Lockdown Period * n (%)				
	Pre-	During 1st	Post-1st	During 2nd	Post-2nd
Number of presentations	1157	141	579	53	329
Mean number of presentations/week	18	9	13	9	12
Patients with BHT data, n (%)	1115 (96.4)	131 (92.9)	576 (99.5)	53 (100)	323 (98.2)
Age in years–Median (IQI)	23 (18, 34)	26 (19, 39)	26 (19, 38)	24 (19, 31)	25 (19, 38)
Sex					
Male	479 (41.4)	66 (46.8)	268 (46.3)	26 (49.1)	143 (43.5)
Female	635 (54.9)	63 (44.7)	310 (53.5)	26 (49.1)	184 (55.9)
Missing	43 (3.7)	12 (8.5)	1 (0.2)	1 (1.9)	2 (0.6)
Time of attendance					
00:00–07:59	130 (11.2)	11 (7.8)	62 (10.7)	2 (3.8)	28 (8.5)
08:00–15:59	354 (30.6)	55 (39.0)	227 (39.2)	22 (41.5)	140 (42.6)
16:00–23:59	628 (54.3)	64 (45.4)	285 (49.2)	28 (52.8)	158 (48)
Missing	45 (3.9)	11 (7.8)	5 (0.9)	1 (1.9)	3 (0.9)
Psychiatric assessment conducted					
Yes	1035 (89.5)	125 (88.7)	546 (94.3)	52 (98.1)	304 (92.4)
No	56 (4.8)	6 (4.3)	25 (4.3)	0 (0)	22 (6.7)
Unknown	66 (5.7)	10 (7.1)	8 (1.4)	1 (1.9)	3 (0.9)
Assessment performed by					
No assessment	56 (4.8)	6 (4.3)	52 (9.0)	3 (5.7)	23 (7.0)
Consultant	42 (3.6)	11 (7.8)	58 (10.0)	2 (3.8)	7 (2.1)
Registrar	403 (34.8)	57 (40.4)	302 (52.2)	34 (64.2)	165 (50.2)
Senior registrar	60 (5.2)	12 (8.5)	9 (1.6)	0 (0)	2 (0.6)
Medical Officer	501 (43.3)	45 (31.9)	148 (25.6)	11 (20.8)	123 (37.4)
Unknown	95 (8.2)	10 (7.1)	10 (1.7)	3 (5.7)	9 (2.7)
Current psychiatric diagnosis					
Yes	533 (46.1)	69 (48.9)	201 (34.7)	18 (34)	119 (36.2)
No	480 (41.5)	53 (37.6)	274 (47.3)	29 (54.7)	155 (47.1)
Unknown	144 (12.4)	19 (13.5)	104 (18.0)	6 (11.3)	55 (16.7)
Method					
Medicine	696 (60.2)	68 (48.2)	351 (60.6)	35 (66)	186 (56.5)
Agrochemical/pesticide/insecticide	225 (19.4)	42 (29.8)	140 (24.2)	12 (22.6)	58 (17.6)
Plant/other	194 (16.8)	21 (14.9)	86 (14.9)	6 (11.3)	84 (25.5)
Missing	42 (3.6)	10 (7.1)	2 (0.3)	0 (0)	1 (0.3)

* Pre-pandemic (1 January 2019–19 March 2020); During 1st lockdown (20 March 2020–27 June 2020); Post-1st lockdown (28 June 2020–11 May 2021); During 2nd lockdown (12 May 2021–21 June 2021); Post-2nd lockdown 22 June 2021–31 December 2021). BHT—Bed Head Ticket. IQI—Interquartile interval.

**Table 2 ijerph-20-01833-t002:** Rate ratios for the difference in the number/rate of admission during the different lockdown periods compared to a pre-pandemic period.

	Lockdown Periods *—Step Change IRR (95% CI)			
	During 1st	Post-1st	During 2nd	Post-2nd
Overall	0.54 (0.44, 0.67)	0.79 (0.63, 0.99)	0.55 (0.37, 0.81)	0.75 (0.53, 1.05)
Sex				
Female	0.45 (0.34, 0.59)	0.79 (0.60, 1.04)	0.51 (0.31, 0.84)	0.79 (0.52, 1.21)
Male	0.61 (0.43, 0.86)	0.88 (0.61, 1.27)	0.65 (0.34, 1.21)	0.78 (0.44, 1.37)
Age group in years				
<25	0.44 (0.32, 0.60)	0.74 (0.54, 1.00)	0.54 (0.32, 0.93)	0.71 (0.44, 1.13)
25+	0.62 (0.46, 0.83)	0.93 (0.68, 1.27)	0.63 (0.36, 1.08)	0.90 (0.55, 1.45)
Methods				
Medicinal	0.45 (0.32, 0.62)	0.85 (0.63, 1.16)	0.66 (0.38, 1.15)	0.76 (0.47, 1.23)
Agrochemical	0.79 (0.53, 1.18)	0.81 (0.51, 1.28)	0.49 (0.22, 1.10)	0.50 (0.24, 1.03)
Plant/other	0.63 (0.40, 0.99)	0.60 (0.37, 0.96)	0.40 (0.15, 1.06)	0.86 (0.43, 1.72)

* Pre-pandemic (1 January 2019–19 March 2020); During 1st lockdown (20 March 2020–27 June 2020); Post-1st lockdown (28 June 2020–11 May 2021); During 2nd lockdown (12 May 2021–21 June 2021); Post-2nd lockdown 22 June 2021–31 December 2021).

**Table 3 ijerph-20-01833-t003:** Basic descriptive data on suicide by pandemic lockdown periods and rate ratios for differences in the rate of suicide during the pandemic compared to a pre-pandemic period.

	Pandemic Period n (%)		
	Pre-	During	IRR (95% CI)
Overall	13,405	5574	0.9 (0.68, 1.19)
Sex			
Female	2797 (20.9)	1093 (19.6)	0.98 (0.84, 1.14)
Male	10,608 (79.1)	4481 (80.4)	0.89 (0.69, 1.16)
Age group in years			
8–25	2563 (19.1)	1057 (19)	0.97 (0.83, 1.13)
26–35	2237 (16.7)	936 (16.8)	0.77 (0.56, 1.06)
36–55	4281 (31.9)	1689 (30.3)	0.94 (0.82, 1.07)
55+	4324 (32.3)	1892 (33.9)	1.05 (0.91, 1.21)
Methods			
Hanging	3421 (25.5)	1091 (19.6)	0.96 (0.75, 1.23)
Poisoning	7636 (57)	3619 (64.9)	0.97 (0.82, 1.15)
Other	2348 (17.5)	864 (15.5)	1.01 (0.75, 1.36)

## Data Availability

Suicide data can be obtained directly from the Police department in Sri Lanka. All other data are available from the authors.

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
