# Peer review of "The Impact of the COVID-19 Pandemic and Lockdowns on Self-Poisoning and Suicide in Sri Lanka: An Interrupted Time Series Analysis"

_ijerph, 2023, doi:10.3390/ijerph20031833_

Round 1

Reviewer 1 Report

1. Please add an explanation of how the model for this study removes the effect of time trends. Has the number of hospital visits for self-poisoning been consistently and naturally decreasing?

2. Please provide model equations in the methods section where appropriate.

3. Is there a robustness test for this study? If not, please add.

4. This study does not compare the differences between low- and middle-income countries and high-income countries when discussing the causes of changes in suicide or self-poisoning. I would have preferred to see different phenomena and perspectives than consistent conclusions.

5. Authors should use their insights as experts, not just complete a data analysis. Please reveal the reasons and mechanisms behind the results of the data as much as possible, rather than responding that it is not clear.

Author Response

We thank the reviewer for their comments and their time reviewing our manuscript. Please see the attachment. 

Reviewer 2 Report

The submitted manuscript is of a study that investigated changes in hospital admission rates for self-poisoning and also suicides during the COVID-19 pandemic. Using an interrupted time series analysis and separate data bases for hospital admissions and suicides the authors found that during COVID lockdowns there was a decrease in self-poisoning admissions and no change in suicides. There were sex and ages differences, but the pattern was similar during lockdowns across groups. The authors also noted the increasing trend in hangings, independent of lockdowns.

Overall, this is a good paper and meaningfully adds to the literature since this study is from a low-middle income nation. A few suggestions for the authors to consider are provided below.

1. Presentation of figures can be improved. The x-axis labeling week is cluttered. If possible try displaying every other label or vertical alignment of labels.

2. The y-axis for figure 1& 4 is number of cases, while for figures 2 & 3 it is cases per 100,000. It would be better to be consistent.

3. Figures 5 and 6 are the exact same. Should one be for sex or age? Please check.  

4. When introducing interrupted time series a reference should be provided to an article explaining the method.

5. It would also be helpful to write the model equation of the ITS model to assist the reader in understanding how it is modeled.

6. It is mentioned that quarantine curfews were implemented in some provinces but not others. Would it be possible to compare provinces on suicides during this period?

Author Response

(The authors gave the same response as above.)
